# Accelerating water dissociation kinetics by isolating cobalt atoms into ruthenium lattice

Junjie Mao[1,2], Chun-Ting He[3], Jiajing Pei[4], Wenxing Chen[2], Dongsheng He[5], Yiqing He[2], Zhongbin Zhuang[4], Chen Chen[2], Qing Peng[2], Dingsheng Wang [2] & Yadong Li[2]

Designing highly active and robust platinum-free catalysts for hydrogen evolution reaction is of vital importance for clean energy applications yet challenging. Here we report highly active and stable cobalt-substituted ruthenium nanosheets for hydrogen evolution, in which cobalt atoms are isolated in ruthenium lattice as revealed by aberration-corrected high-resolution transmission electron microscopy and X-ray absorption fine structure measurement. Impressively, the cobalt-substituted ruthenium nanosheets only need an extremely low overpotential of 13 mV to achieve a current density of 10 mA cm$^{-2}$ in 1 M KOH media and an ultralow Tafel slope of 29 mV dec$^{-1}$, which exhibit top-level catalytic activity among all reported platinum-free electrocatalysts. The theoretical calculations reveal that the energy barrier of water dissociation can greatly reduce after single cobalt atom substitution, leading to its superior hydrogen evolution performance. This study provides a new insight into the development of highly efficient platinum-free hydrogen evolution catalysts.

[1] Key Laboratory of Functional Molecular Solids, Ministry of Education, Anhui Key Laboratory of Molecule—Based Materials, College of Chemistry and Materials Science, Anhui Normal University, Wuhu 241000, China. [2] Department of Chemistry, Tsinghua University, Beijing 100084, China. [3] MOE Key Laboratory of Functional Small Organic Molecule, College of Chemistry and Chemical Engineering, Jiangxi Normal University, Nanchang 330022, China. [4] State Key Lab of Organic-Inorganic Composites, Beijing University of Chemical Technology, Beijing 100029, China. [5] Materials Characterization and Preparation Center (MCPC), South University of Science and Technology of China, Shenzhen, Guangdong 518055, China. These authors contributed equally: Junjie Mao, Chun-Ting He, Jiajing Pei. Correspondence and requests for materials should be addressed to D.W. (email: wangdingsheng@mail.tsinghua.edu.cn)

The electrocatalytic hydrogen evolution, as an attractive strategy to the production of hydrogen, has been receiving great attention in clean alternative energy technologies[1-5]. Generally, the hydrogen evolution reaction (HER) in acidic media is much easier than that in alkaline media. This is because the reaction precursor in acidic media is a proton which can easily receive electrons from the cathode. The sluggish kinetics of the HER, particularly in alkaline electrolytes, calls for developing highly efficient catalysts to lower the hydrogen evolution overpotential. To date, Pt is the most popular HER catalyst due to its high exchange current density ($j_0$) and small overpotentials ($\eta$). However, the scarcity, high price and poor durability of Pt greatly restrict its commercial applications. To address these issues, many research groups have employed various approaches to obtaining Pt-free catalysts with superior HER catalytic activity. For instance, constructing unique nanostructures (nanoframes[6,7], defect-rich nanosheets/nanowires[8-11], porous/hybrid nanostructures[12-19], etc.) with abundant active sites; introducing heteroatoms like N, P, S into metal elements to form metal-N/P/$S_x$ catalysts[15,20-22]; preparing single atom catalysts supported on porous carbon nanomaterials[23-25]; loading catalysts on a proper substrate/support[23,24,26-28] and so on. Although much work has been done in this field, only a few of the instances of Pt-free catalysts with optimized structure and composition have shown the satisfactory activity in comparison with Pt. On the other hand, the irregular structure and/or complex compositions of the reported catalysts make it difficult to understand the relationship between catalytic sites at the atomic scale and hydrogen evolution performance. Therefore, it is necessary to develop new strategies for designing novel Pt-free electrocatalysts at the atomic level, with much enhanced catalytic activity and durability for HER than commercial Pt/C catalyst.

In this study, we prepare Co-substituted Ru nanosheets (NSs) with atomically dispersed Co into Ru lattice. Remarkably, the as-obtained Co-substituted Ru NSs exhibit superior catalytic activity towards HER in 1 M KOH media, with an ultralow overpotential of 13 mV to achieve a current density of 10 mA cm$^{-2}$ and an ultralow Tafel slope of 29 mV dec$^{-1}$, and excellent durability over 1000 cycles. To our knowledge, the Co-substituted Ru NSs exhibit the top-level catalytic activity for HER among all reported Pt-free electrocatalysts. The experimental results and density functional theory (DFT) calculations further demonstrated that single Co atom substituted Ru catalysts can remarkably reduce the energy barrier of water dissociation in comparison with pristine Ru and RuCo alloy, thus leading to superior performance toward HER.

## Results

**Synthesis and characterizations**. In a typical synthesis, Co-substituted Ru NSs were synthesized by co-reduction of Ru(acac)$_2$ and Co(acac)$_2$ in the mixed solutions containing oleylamine and heptanol at 180 °C for 12 h. Transmission electron microscopy (TEM) and high-angle annular dark-field scanning TEM (HAADF-STEM) were conducted to investigate the morphology and structure of the as-obtained products. As shown in Fig. 1a and Supplementary Fig. 1, the uniform hexagon NSs with a lateral size around 30 nm were synthesized. Interestingly, the obtained nanosheets can self-assemble into patterned arrays (Supplementary Fig. 2) when depositing high concentration of nanosheets on the copper grid. Atomic force microscopy was utilized to determine the thickness of NSs (Fig. 1b). Section analysis and height profile show that the thickness of the obtained NSs is about ~1.6 nm (~7 atomic layers of Ru). The atomic structure of nanosheets is further examined by atomic resolution aberration-corrected HAADF-STEM imaging technique. Figure 1c shows that the obtained NSs have the single-crystalline nanostructure, which possesses the side plane of Ru (01$\bar{1}$0) facets, and the basal plane of Ru (0001) facets. Moreover, the enlarged image of the surfaces of nanosheet also reveals that the Co atoms locate at the hexagonal close-packed (hcp) Ru lattice positions. Figure 1d shows the HAADF line scanning profile of Co-substituted Ru NSs. Figure 1e

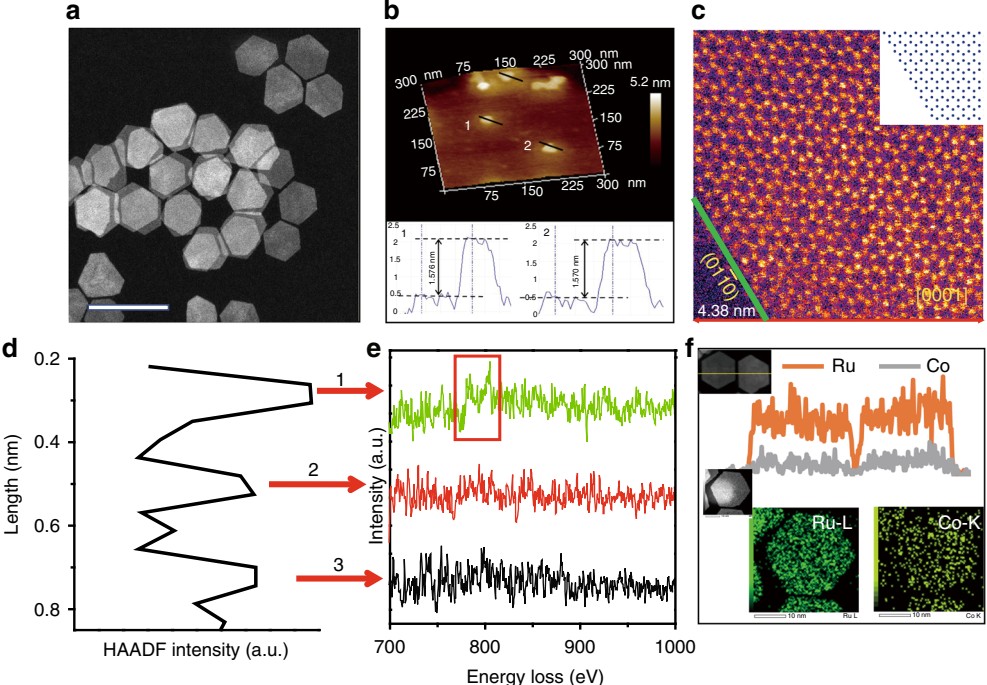

**Fig. 1** Characterization of Co-substituted Ru NSs. Representative **a** STEM image. Scar bar: 50 nm. **b** AFM image. **c** Atomic resolution HAADF-STEM image of a single Co-substituted Ru nanosheet. False color was used to increase the contrast. Inset of **c** is the surface model of the nanosheet. **d** Line scanning profile and **e** background subtracted EELS spectrum of the Co-substituted Ru NSs acquired from **d**. The red rectangle represents Co L$_{2,3}$-edge signal. **f** EDX line scanning profile and EDX mapping of a single Co-substituted Ru NS

shows the simultaneously acquired electron energy loss spectroscopy (EELS) spectra of the Co K-edge taken from the three atomic column locations indicated in Fig. 1d. We only found that position 1 exhibits typical Co $L_{2, 3}$-edge signal, while we could not observe the similar signals from neighboring atoms, indicating a highly monodispersed distribution of Co atoms in the NSs. The X-ray diffraction (XRD) pattern (Supplementary Fig. 3) of the NSs could be assigned to (10$\bar{1}$0), (0002) and (10$\bar{1}$1) diffractions of a hcp structure. No obvious diffraction peak shift can be found due to the low content of Co. As a comparison, the diffraction pattern of RuCo alloy (Supplementary Fig. 4) is slightly shifted to higher angles compared to those of pristine Ru, corresponding to the lattice contraction resulting from the incorporation of Co atoms into Ru nanosheets. To better understand the elemental distribution of Ru and Co elements, the energy-dispersive X-ray (EDX) spectroscopy mapping and the EDX line scanning profile were collected. As shown in Fig. 1f, Ru and Co elements were distributed homogeneously throughout the structure. EDX analysis further demonstrates that the NSs are composed of Ru and Co elements with atomic ratios of 94 and 6% (Supplementary Fig. 5). The atomic ratio of Ru and Co was higher than that of the reactants, which was similar to our previous studies[29].

To precisely investigate the local atomic and electronic structures of the Co-substituted Ru samples, extended X-ray absorption fine structure (EXAFS) and X-ray absorption near-edge structure (XANES) measurements were conducted. Figure 2a shows the XANES curves at Ru K-edge of Co-substituted Ru NSs. It is found that the Ru K-edge of the sample slightly shifts to lower energy in comparison to that of Ru foil reference, suggesting that Ru in Co-substituted Ru NSs gets some electrons from Co. From the XANES spectra of Co K-edge in Fig. 2b, we found that the absorption threshold of Co-substituted Ru NSs is between those of Co foil and Co-O, indicating that Co in Co-substituted Ru NSs is positive charged. In order to further investigate the interaction between Ru and Co, Fourier transform

(FT) EXAFS analysis is performed at Ru K-edge (Fig. 2c) and Co K-edge (Fig. 2d), respectively. In FT-EXAFS spectra of Ru K-edge, there is no peak related to Ru-O scattering, and the peak at 2.39 Å (slightly shorter than 2.41 Å of Ru–Ru in Ru foil) is due to Ru–Ru/Co contribution. In FT-EXAFS curves of Co K-edge (Fig. 2d), Co-O coordination is not observed and the peak at 2.39 Å results from Co–Ru interaction. It is necessary to mention that the position of Co–Ru peaks is obviously positive than that of Co–Co in Co foil (2.16 Å). This result indicates the absence of Co–Co scattering in Co-substituted Ru NSs and demonstrates the monodispersing of Co atoms in Co-substituted Ru NSs.

**Catalytic properties.** To demonstrate the superior catalytic properties, the electrocatalytic activities of the as-obtained Co-substituted Ru NSs toward HER were investigated in hydrogen-saturated 1 M KOH solution using a three-electrode system (see Supporting Information for details, Supplementary Fig. 6-8). As a comparison, commercial Pt/C, Ru/C, RuCo and $RuCo_2$ alloy (Supplementary Fig. 9, 10) were measured under the same conditions. Figure 3a shows the linear sweep voltammetry (LSV) curves for the HER over the Co-substituted Ru, RuCo alloy, Ru/C and Pt/C catalysts. As can be seen from Fig. 3a and Supplementary Fig. 10a, the Co-substituted Ru catalyst shows the highest catalytic activity than that of $RuCo_2$, RuCo, Ru/C and Pt/C catalysts. This result indicates that the monatomic substitution of Co can greatly enhance the catalytic activity of HER, while the presence of Co–Co bond in RuCo and $RuCo_2$ alloy would lead to the decrease of the catalytic activity. Strikingly, the Co-substituted Ru NSs only required an overpotential as low as 13.0 mV to achieve current density of 10 mA cm$^{-2}$, which is much lower than that of $RuCo_2$ (144 mV), RuCo (40.0 mV), Ru/C (92.5 mV) and Pt/C (56.5 mV) catalysts (Fig. 3b) and many other Pt-free catalysts (Supplementary Table 1), suggesting its superior HER activity. The HER kinetics of the above catalysts were also calculated via corresponding Tafel plots. As shown in Fig. 3c and

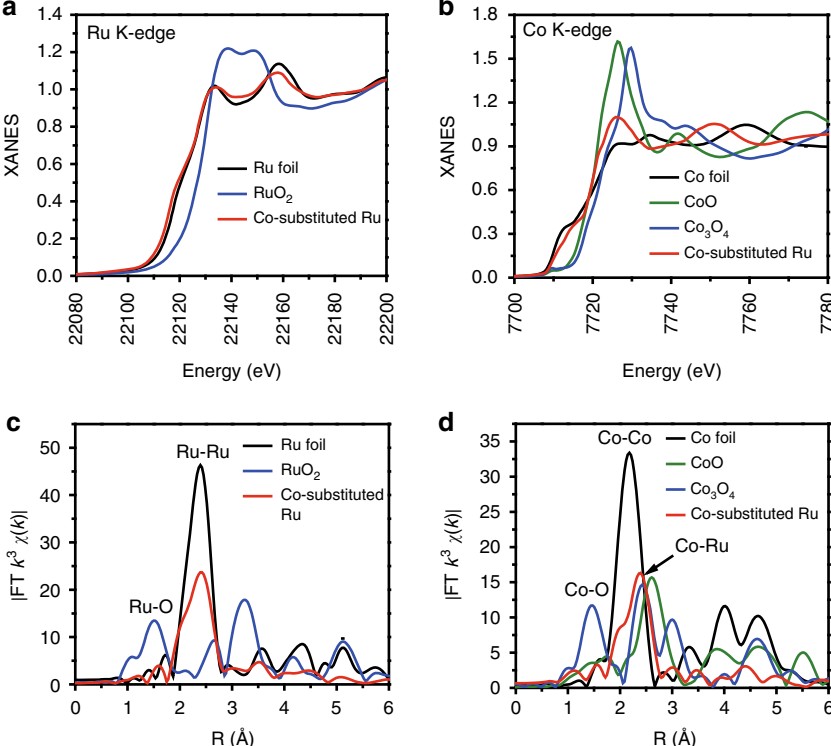

**Fig. 2** Structure determination of Co-substituted Ru by XAFS. XANES spectra of Co-substituted Ru for **a** Ru K-edge and **b** Co K-edge. Fourier transformed (FT) $k^3$-weighted $\chi(k)$-function of the EXAFS spectra for **c** Ru K-edge and **d** Co K-edge

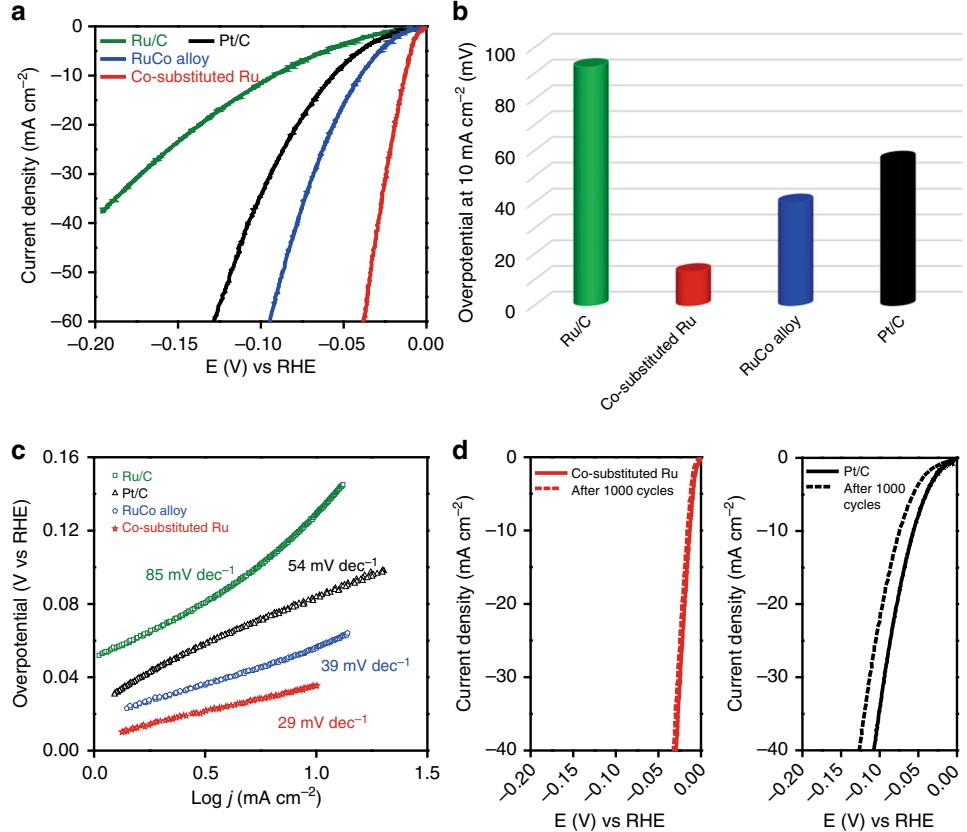

**Fig. 3** Comparison of catalytic ability of Ru/C, Pt/C, RuCo alloy and Co-substituted Ru. **a** LSV curves, scan rate: 5 mV s$^{-1}$. **b** Overpotentials at 10 mA cm$^{-2}$ of Ru/C, Pt/C, RuCo and Co-substituted Ru. **c** Tafel plots of Ru/C, Pt/C, RuCo and Co-substituted Ru in 1 M KOH. **d** Durability test of Pt/C and Co-substituted Ru, the LSV curves measured before and after 1000 cycles at a scan rate of 5 mV s$^{-1}$

Supplementary Fig. 10b, the Tafel slope of the Co-substituted Ru is 29 mV dec$^{-1}$, which is much smaller than that of Ru/C (85 mV dec$^{-1}$), Pt/C (54 mV dec$^{-1}$), RuCo$_2$ (122 mV dec$^{-1}$) and RuCo alloy (39 mV dec$^{-1}$), demonstrating its more efficient kinetics of hydrogen evolution. We estimated the turnover frequency (TOF) of the above catalysts to further investigate their intrinsic HER activities (Supplementary Fig. 11, 12). The TOF values of Co-substituted Ru at 30 and 60 mV are 2.15 H$_2$ s$^{-1}$ and 6.39 H$_2$ s$^{-1}$, respectively, both of which are much higher than those of Pt/C (0.43 H$_2$ s$^{-1}$ at 30 mV; 1.69 H$_2$ s$^{-1}$ at 60 mV), Ru/C (0.41 H$_2$ s$^{-1}$ at 30 mV; 1.23 H$_2$ s$^{-1}$ at 60 mV) and RuCo alloy (0.51 H$_2$ s$^{-1}$ at 30 mV; 2.07 H$_2$ s$^{-1}$ at 60 mV), further revealing that Co-substituted Ru NSs have a superior intrinsic HER activity.

To evaluate the durability of the Co-substituted Ru NSs, long-term cyclic voltammetry between 0 V and −0.20 V (vs RHE) was examined. The commercial Pt/C catalyst was used as a reference under the same conditions. As shown in Fig. 3d, the polarization curve of Pt/C, negatively shifted about 47 mV after 1000 cycles at a current density of 10 mA cm$^{-2}$, indicating an obvious decrease in the electrocatalytic activity. As for Co-substituted Ru NSs, the polarization curve of Co-substituted Ru after 1000 continuous cycles was almost overlapped with the initial one, suggesting the durability of Co-substituted Ru NSs under the electrocatalytic processes. The durability of the Co-substituted Ru NSs was further evaluated by chronopotentiometry at a constant current density of 10 mA cm$^{-2}$ in 1.0 M KOH electrolytes. As shown in Supplementary Fig. 13, the Co-substituted Ru catalyst shows negligible degradation in the course of 72,000 s. The TEM image (Supplementary Fig. 14) and EDX results (Supplementary Fig. 15) show no obvious morphology and composition change for Co-substituted Ru catalyst even after 1000 cycle durability test. Furthermore, we

utilize the in situ XAFS measurements (Fig. 4a) to monitor the atomic structure evolution of Ru and Co in Co-substituted Ru catalyst during the catalytic reaction process. As shown in Fig. 4b–e, no obvious change can be found from the in situ XANES spectra and EXAFS spectra for the Ru K-edge and Co K-edge at different voltages, demonstrating the high stability of the local atomic structure of Co-substituted Ru catalyst. Therefore, introducing monatomic Co into Ru lattice can greatly boost the electrocatalytic activity and durability toward HER, making it much more active than Pt/C catalyst and even all the Pt-free catalysts.

**DFT studies on Co-substituted Ru catalysts**. We further performed DFT calculations to gain insights into the atomic Co substitution effect on the HER activity. Generally, the typical process of HER (2H$^+$+2e$^-$→H$_2$) in alkaline condition is a two-step electrochemical process taking place on the catalytic active center that finally generates gaseous hydrogen[30]. First is the electrochemical hydrogen adsorption (Volmer step; Eq. (1))

$$H_2O + M + e^- \rightarrow M-H^* + OH^-, \qquad (1)$$

followed by the electrochemical desorption (Heyrovsky step; Eq. (2)) or chemical desorption (Tafel step; Eq. (3))

$$M-H^* + H_2O + e^- \rightarrow M + H_2 + OH^-, \qquad (2)$$

$$2M-H^* \rightarrow 2M + H_2. \qquad (3)$$

Obviously, the chemical adsorption and desorption of H atoms on the catalyst surface should be neither too strong nor too weak as they are competitive processes. According to the Sabatier[31] principle, in the viewpoint of thermodynamics, a good HER

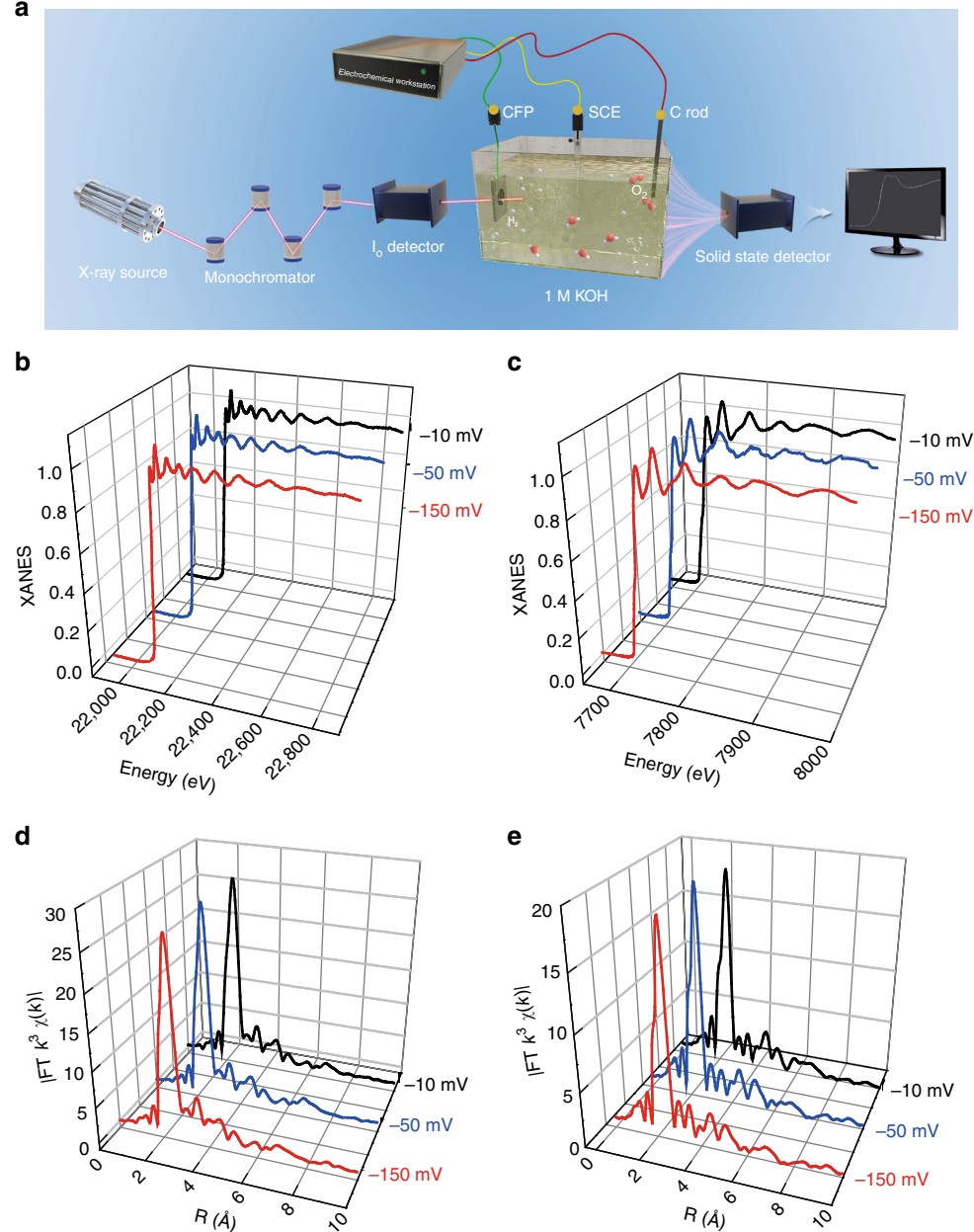

**Fig. 4** In situ XAFS measurements for Co-substituted Ru catalyst. **a** Schematic illustration of in situ XAS measurements for Co-substituted Ru. CFP and SCE represent carbon fiber paper and saturated calomel electrode, respectively. XANES spectra of Co-substituted Ru for the **b** Ru K-edge and **c** Co K-edge at different voltages. EXAFS spectra for the **d** Ru K-edge and **e** Co K-edge at different voltages

catalyst should possess a hydrogen adsorption-free energy ($\Delta G_{H^*}$ as an activity descriptor) with an optimal value around zero[32,33]. Thus, we first calculated the hydrogen adsorption-free energy on Ru (0001) surfaces with or without Co substitution. The results showed that the $\Delta G_{H^*}$ values after Co atom substitution are more negative than that of pristine Ru (Fig. 5a), indicating stronger hydrogen binding abilities after Co atom substitution, which should lead to the obstacle of thermodynamic H* desorption process in the Heyrovsky step. Actually, relatively inert $H_2O$ molecules are the reacting species for HER in alkaline media, and hence the water dissociation kinetic of Volmer step, namely the energy barrier ($\Delta G_w$) of O–H bond cleavage, is usually more crucial to the overall reaction rate[34]. As shown in Fig. 5b, the favorable hydrogen desorption process is the Tafel step with $\Delta G_w$ value of 13.42 kcal mol$^{-1}$, being much lower than that of Heyrovsky step ($\Delta G_w = 22.47$ kcal mol$^{-1}$). Moreover, it is also

obviously lower than that of the Volmer step ($\Delta G_w = 19.28$ kcal mol$^{-1}$), indicating that the water dissociation is indeed the rate-limiting process of HER in this material. Expectedly, the energy barrier of water dissociation has remarkably reduced after single Co atom substitution (denoted as RuCo1) compared with that of pristine Ru ($\Delta G_w = 26.48$ kcal mol$^{-1}$), being consistent with the experimental observation. However, when increasing the substituted Co atoms to two (denoted as RuCo2) and three (denoted as RuCo3) per unit cell, the energy barriers instead increase to 23.01 and 21.89 kcal mol$^{-1}$ (Fig. 5a, c), respectively, meaning that more Co substitution can lead to sluggish water dissociation due to the destruction of single atom dispersion. Moreover, the OH binding energy ($\Delta E_{OH}$) gradually increases with the addition of Co atoms (Fig. 5a), which should result from the increasing electron density in the saddle points (groove formed by the three nearest atoms) around Co atom (Fig. 5d). This change, on one

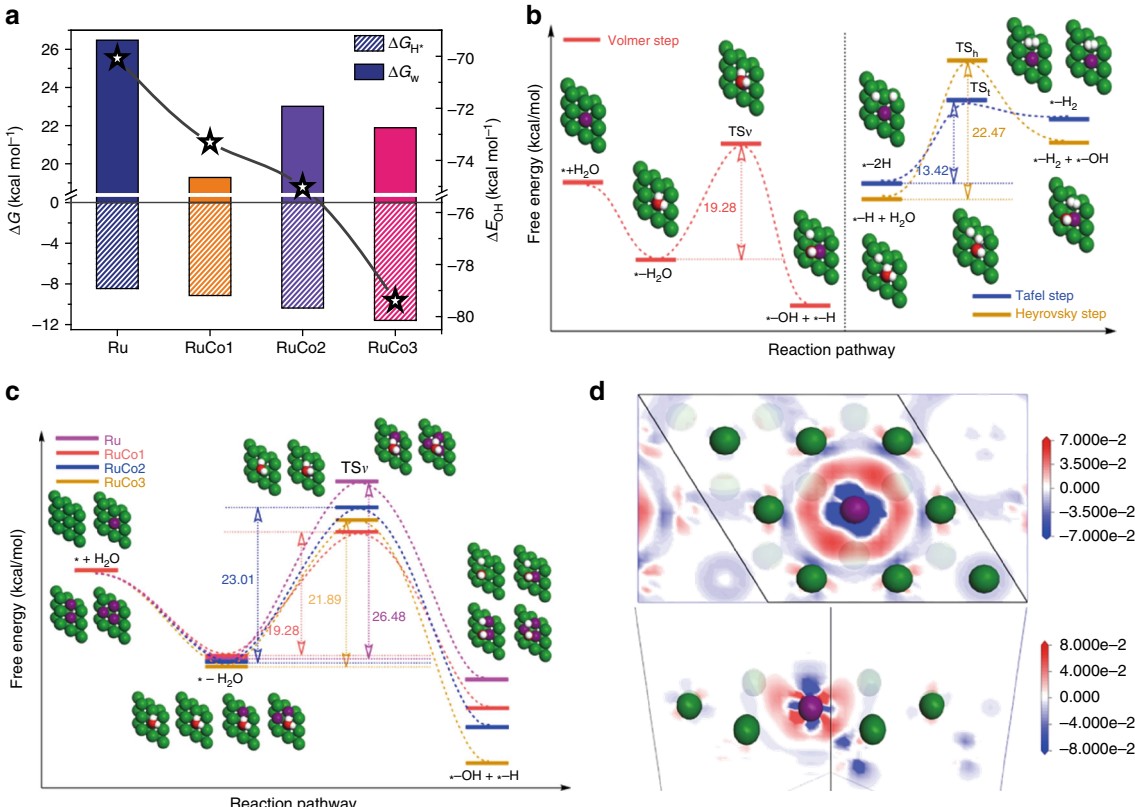

**Fig. 5** DFT calculations to gain insights into the atomic Co substitution effect on the HER activity. **a** The hydrogen adsorption-free energy ($\Delta G_{H^*}$), water dissociation energy barrier ($\Delta G_w$) and OH binding energy ($\Delta E_{OH}$) on various metal surfaces with different amount of Co substitution. **b** Free energy diagrams of the three elementary processes of HER on RuCo1 including atomic configurations of reactant initial states, intermediate state, final states and additional transition states (TSv, TSh and TSt represent the transition states of Volmer, Heyrovsky and Tafel step, respectively). **c** Free energy diagrams of the Volmer steps of HER on various metal surfaces with different amounts of Co substitution including atomic configurations of reactant initial states, intermediate state, final states and additional transition states (TSv). **d** Top view (up) and side view (down) of the electron density change plots after single Co atom substitution. Color: green, Ru; purple, Co; red, O; white, H

hand, benefits for boosting the cleavage of O–H bond, but on the other hand, causes adverse OH to be desorbed and then blocks the consequent reactions. Therefore, a moderate $\Delta E_{OH}$ value of the single atom doped structure can keep a favorable balance between facilitating water dissociation and preventing the active sites from inhibiting by the high-affinity hydroxides and finally accelerates reaction kinetics.

## Discussion

In summary, Co-substituted Ru NSs as highly efficient Pt-free HER catalysts were successfully synthesized. The structure of Co-substituted Ru NSs was characterized by aberration-corrected HAADF-STEM and XAFS measurements, indicating the atomically dispersed Co into Ru lattice. More importantly, the Co-substituted Ru NSs exhibit superior catalytic performance for HER in alkaline, which is much better than the activity of commercial Pt/C catalyst and other Pt-free catalysts. Theoretical calculations suggested that single Co atom substitution significantly reduce energy barrier of water dissociation, leading to the outstanding HER performance. Our findings not only present new opportunities to design highly active HER catalysts at atomic scale, but also advance our understanding of HER through theoretical insights.

## Methods

**Characterization**. The powder XRD patterns were recorded with a Bruker D8-advance X-ray powder diffractometer with monochromatized CuKα radiation ($\lambda = 1.5406$ Å). The morphology and structure of the obtained products were determined using a Hitachi model H-800 TEM and a FEI Tecnai G2 F20 S-Twin high-resolution TEM operated at 200 kV. The HAADF-STEM images were performed using a Titan 80–300 scanning/transmission electron microscope operated at 300 kV, equipped with a probe spherical aberration corrector.

**Preparation of Co-substituted Ru and RuCo alloy**. In a typical synthesis of Co-substituted Ru nanosheets, 40 mg of $Ru_3(CO)_{12}$, 5.2 mg of $Co(acac)_2$, 500 mg of glucose and 100 mg of citric acid monohydrate were dissolved in 20 mL of oleylamine and 10 mL of 1-heptanol mixed solvents, followed by vigorous stirring for 2 h. Then, the resulting homogeneous solution was transferred into a 50 mL Teflon-lined autoclave and heated at 180 °C for 12 h. After cooling down to the room temperature, the black precipitate was obtained by centrifugation, and washed with the ethanol-cyclohexane mixture for several times. Finally, the Co-substituted Ru nanosheets were obtained. The preparation of RuCo alloy was the same as Co-substituted Ru nanosheets except that the amount of Co was changed to 48 mg.

**XAFS measurements and analysis**. The XAFS (Ru K-edge and Co K-edge) spectra were collected at BL14W1 station in Shanghai Synchrotron Radiation Facility (SSRF, the storage rings were operated at 3.5 GeV with a maximum current of 250 mA). The data were collected at room temperature in transmission mode (Si(311) monochromator for Ru K-edge and Si(111) monochromator for Co K-edge). All samples were pelletized as disks of 13 mm diameter using graphite powder as a binder (ground thoroughly by mortar and pestle). The acquired EXAFS data were processed according to the standard procedures using the ATHENA module implemented in the IFEFFIT software packages. The EXAFS spectra were obtained by subtracting the post-edge background from the overall absorption and then normalizing with respect to the edge-jump step. Subsequently, $\chi(k)$ data in the $k$-space were Fourier transformed to real (R) space using a hanning windows (dk = 1.0 Å$^{-1}$) to separate the EXAFS contributions from different coordination shells.

**Electrochemical measurements**. The electrochemical measurements were conducted on a three-electrode system (Supplementary Fig. 6) controlled by a potentiostat (V3, Princeton Applied Research). The catalyst-modified glassy carbon (GC) electrodes, graphite rod and saturated calomel electrode (SCE) were used as working electrode (geometric area of a working electrode: 0.196 cm$^2$), counter electrode (CE) and reference electrode (the SCE was calibrated, Supplementary Fig. 7), respectively. Before test, GC substrate electrodes were polished with 0.3 and 0.05 mm Al$_2$O$_3$ slurry and then sonicated in ethanol and water each for several times to make it clean. For preparation of the electrode materials, 2 mg of catalyst powder was dispersed in isopropanol/distilled water (Milli-Q, volume ratio, 1:1) mixed solution containing 0.05 wt% of Nafion. The mixture was ultrasonicated for about 2 h to generate a homogeneous ink. After that, 15 μL of ink solutions was transferred onto the GC rotating disk electrode and then dried at room temperature. We optimize the catalytic activity of the Co-substituted Ru catalysts by tuning the loading mass before electrochemical test. As shown in Supplementary Fig. 8, the optimal loading of the catalysts was 0.153 mg cm$^{-2}$ in 1.0 M KOH. The hydrogen-saturated 1 M KOH aqueous solutions were used as the electrolytes before each test. The scan rate was set to 5 mV s$^{-1}$ for LSV measurements. All polarization curves were corrected for the iR compensation (the specific percentage of the correction is 100%). The electrochemical tests were performed at room temperature.

The TOF values can be calculated by the following equation: TOF $= I/(2Fn)$, where $I$ represents the current, $F$ represents the Faraday constant and $n$ represents the number of active sites. Forming H$_2$ needs two electrons that leads to the factor of 1/2 in this equation. Here, the number of active sites can be qualified by the following equation[19,35–37]: $n = Q_{Cu}/2F$, where $Q_{Cu}$ represents the underpotential deposition (UPD, Supplementary Fig. 11) copper stripping charge (Cu upd→Cu$^{2+}$ + 2e$^-$).

**Details of DFT calculations**. The spin polarization DFT calculations were performed by the Dmol$^3$ module in Materials Studio 5.5 package and generalized gradient approximation with Perdew–Becke–Ernzerhof (PBE) was used for the exchange–correlation functional. The double numerical plus polarization (DNP) basis set were adopted, while an accurate DFT semi-core pseudopots (DSPP) was employed for the metal atoms. All of the models are calculated in periodically boxes with a vacuum slab of 15 Å to separate the interaction between periodic images. The simulated unit cell is hexagonal with 8.12 × 8.12 × 19.28 Å$^3$. The transition state search was performing with a linear synchronous transit (LST), followed by repeated conjugate gradient minimizations and quadratic synchronous transit (QST) maximizations until a transition state has been located. All the transition state configurations were confirmed through the frequency analysis. The energy, gradient and displacement convergence criteria were set as $1 × 10^{-5}$ Ha, $2 × 10^{-3}$ Å and $5 × 10^{-3}$ Å, respectively. The Gibbs free energy of each elementary step was calculated as

$$\Delta G = \Delta E + \Delta ZPE - T\Delta S, \qquad (4)$$

where $\Delta E$ is the reaction energy calculated using the spin polarization DFT method. $\Delta ZPE$ and $\Delta S$ are the changes in zero-point energies and entropy during the reaction, respectively. Particularly, as the vibrational entropy of H* in the adsorbed state is small, the entropy of adsorption of 1/2 H$_2$ is $\Delta S_H \approx -0.5S_{0H2}$, where $S_{0H2}$ is the entropy of H$_2$ in the gas phase at the standard conditions.

## Data availability

The data supporting this study are available from the authors on reasonable request.

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

## Acknowledgements

This work was supported by the China Ministry of Science and Technology under Contract of 2016YFA (0202801), the National Natural Science Foundation of China (21521091, 21390393, U1463202, 21471089, 21671117) and the Beijing Natural Science Foundation (2174081). We thank the BL14W1 in SSRF, and BL10B and BL12B in NSRL for XAS measurement.

## Author contributions

D.W., Y.L. and J.M. conceived the idea for the project. J.M., J.P. and Y.H. conducted material synthesis. D.H. and J.P. performed structural characterizations and catalytic test. W.C., J.P. and Z.Z. conducted XAFS measurement and analyzed data. C.H. performed DFT calculations. J.M., J.P., D.W., C.C. and Q.P. discussed the catalytic results. D.W., J.M., C.H., W.C. and Y.L. drafted the manuscript. J.M., C.H. and J.P. contributed equally to this work. All authors discussed and commented on the manuscript.

## Additional information

**Competing interests:** The authors declare no competing interests.

