## [Peer Review File · Nature Communications]

Reviewers' comments:

Reviewer #1 (Remarks to the Author):

Recommendation: publish after minor revision

The use of bimetallic Co-Ru alloys for efficient HER is not absolutely new but this paper proposes an original approach and promising results. The paper shows that low concentrations of Co atoms on the surface of a Ru crystal, create isolated Co centers acting as catalytic sites with high activity in the HER from alkaline solutions.

The subject is interesting and appropriate to the Journal. The experiments are sound, conclusions appear well sustained by experimental evidence and DFT calculations. The text is clear and the English style adequate.

As the only objection, I find that the bibliography is limited to very recent publications. Concerning e.g. the DFT calculations, a classical paper of the Norskov group should be mentioned (E. Skulason et al., Modeling the Electrochemical Hydrogen Oxidation and Evolution Reactions on the Basis of Density Functional Theory Calculations, J. Phys. Chem. C 114, 18182-18197 (2010).

Reviewer #2 (Remarks to the Author):

The authors report in this manuscript the HER performance of Co-substituted Ru nanosheets. They have carried out comprehensive structural characterization and confirmed that Co atoms are separate with each other when substituting Ru in the lattice. Impressively, with Co-substituted Ru nanosheets they achieved outstanding catalytic performance for HER in alkaline media. Furthermore, they performed DFT calculations supporting that Co-substitution indeed expedites water dissociation kinetics. Overall, I think this is an interesting and important work, and I would like to recommend the Editor accept the manuscript for publication after revisions as noted below.

1. Ru clusters have been recently reported to outperform Pt for HER in alkaline media. Some key recent progress should be included as references in the manuscript, e.g. Nature Nanotechnology, 2017, 12, 441; Energy Environ. Sci. 2018, 11, 1819
2. In the Introduction section, the authors mention several strategies for improving the HER performance (lines 41-45). However, they miss an important one: the substrate/support effect. In some cases, loading catalysts on a proper substrate/support can alter the electronic structure of catalysts and thereby substantially boost the catalytic performance, like that demonstrated in the above EES paper.
3. The overpotential needed to deliver a current density of 10 mA cm⁻² is only an indicator of apparent catalytic activity and is heavily dependent on the catalyst loading. The authors are recommended to specify the loading mass of Co-substituted Ru NSs and all other control catalysts. It is unfair if these catalysts are compared with different loading mass. Alternatively, the authors may consider optimizing the catalytic activity by tuning the loading mass, and making comparison based on the optimal activity. Furthermore, they need to elaborate how the TOF values are calculated either in the main text or in the Methods section. Are they calculated based on the catalyst mass or ECSA?

4. The authors are recommended to provide chronopotentiometry data to demonstrate long-term stability of their catalysts.

5. There are some technical improvements that should be made, e.g.

Figure 1 caption: the authors should mention what the white circles indicate

Figure 5: the sub-panels should be presented in alphabetical order, namely, a, b, c, d, instead of a, b, d, c

Figures S5 and S8: No signals are visible from Co. Perhaps the authors need to provide a zoomed view of the spectrum. In principle, if the atomic ratio of Co is above 10%, Co K alpha and L alpha should have been observed.

Line 87, page 2: it should be "diffraction pattern", but not "diffraction spectra". "spectrum" usually refers to the variation of intensity as a function of wavelength/energy.

Line 99, page 2: it is better to insert "measurements" after "(XANES)"

The authors should specify how much iR compensation they have made.

6. There are many grammatical errors and typos that should be corrected, to just name a few

a) lines 23-24, abstract: it will read better if the sentence changes to "among all Pt-free electrocatalysts reported so far"

b) Line 27, abstract: it should be "new insight into the development of" or "new insight into developing"

line 39, page 1: it is better to replace "meet" with "address"

line 40, page 1: it should be "approaches to obtaining"

line 50, page 1: delete "very"

line 87, page 2: "are" should be corrected to "is". There is only one pattern in Figure S4.

Line 91, page 2: it is better to move "profile" to a place between "scanning" and "were"

Line 93, page 2: delete "Energy ...spectroscopy". It's already mentioned in line 90. "demonstrate" should be corrected to "demonstrates"

Line 112, page 2: "position" should be used together with "positive" or "negative", and "longer" should be used together with "distance" or "length"

Line 113, page 3: "highest" cannot be used together with "than"

Line 128, page 3: I would suggest replacing "all" with "many other", unless the authors can really compare Co-substituted Ru with ALL Pt-free HER catalysts. This is also the case in line 158.

Line 147, page 3: replace "to" with "with"

Lines 178-179: not clear what this sentence means

Line 185: replace "However" with "Moreover"?

Line 190: "increase" should be corrected to "increasing"

Line 251: "electrodes" should be corrected to "electrode"

Reviewer #3 (Remarks to the Author):

The main claim of this work is the integration of cobalt into ruthenium lattice as single atoms. This structure resulted in high HER activity in alkaline solution. In general, the performance is very good, but there are concerns of the claim on single Co atom in Ru sheets. Major revisions on single atom analysis are recommended.

1. Synthesis: It is doubtful the synthesis procedure could achieve fine single atom catalysts, particularly that cobalt could be reduced and bound to ruthenium in the nanosheets using oleylamine/heptanol at 180 degree C. The added molar ratio 3:1 (Ru:Co) was high and the EDX also suggests a high cobalt amount of 11%. Based on the Fig. 1b and 1c, what should be the estimated

Ru:Co ratio in the catalyst?

2. HAADF-STEM: It is hard to conclude the atoms circled in white are cobalt.

3. RuCo-alloy: Show TEM and EDX data for the comparison sample with the Ru:Co molar feed ratio of 1:3. There was not enough comparisons between the alloy and the single atom catalysts. To a certain extent, having higher concentrations of cobalt with comparable activity is potentially an improvement.

4. In the Co K-edge FT EXAFS analysis, the Co-Ru fit is not convincing enough. There are other possible fits around $R \sim 2-2.7$ for the main peak (with a shoulder peak) of the red curve in Fig. 2.

5. The details of electrochemical measurement should be added, including the loading amount of catalyst and scan rate. How long was the durability test?

Responses to the Referees' Comments

We thank the referees for their valuable comments and positive endorsement to our manuscript. We have carefully considered the referees' comments and revised the manuscript accordingly. Our responses and corresponding revisions are as follows:

Response to Reviewer 1:

Comment 1: Recommendation: publish after minor revision.

The use of bimetallic Co-Ru alloys for efficient HER is not absolutely new but this paper proposes an original approach and promising results. The paper shows that low concentrations of Co atoms on the surface of a Ru crystal, create isolated Co centers acting as catalytic sites with high activity in the HER from alkaline solutions. The subject is interesting and appropriate to the Journal. The experiments are sound, conclusions appear well sustained by experimental evidence and DFT calculations. The text is clear and the English style adequate.

As the only objection, I find that the bibliography is limited to very recent publications. Concerning e.g. the DFT calculations, a classical paper of the Norskov group should be mentioned (E. Skulason et al., Modeling the Electrochemical Hydrogen Oxidation and Evolution Reactions on the Basis of Density Functional Theory Calculations, J. Phys. Chem. C 114, 18182-18197 (2010).

Response: Thank you for your positive comments on our manuscript. We have added the reference you suggested and other references at the right place.

Response to Reviewer 2:

The authors report in this manuscript the HER performance of Co-substituted Ru nanosheets. They have carried out comprehensive structural characterization and confirmed that Co atoms are separate with each other when substituting Ru in the lattice. Impressively, with Co-substituted Ru nanosheets they achieved outstanding catalytic performance for

HER in alkaline media. Furthermore, they performed DFT calculations supporting that Co-substitution indeed expedites water dissociation kinetics. Overall, I think this is an interesting and important work, and I would like to recommend the Editor accept the manuscript for publication after revisions as noted below.

Response: Thank you for your positive comments on our manuscript. We have revised our manuscript accordingly.

Comment 1: Ru clusters have been recently reported to outperform Pt for HER in alkaline media. Some key recent progress should be included as references in the manuscript, e.g. Nature Nanotechnology, 2017, 12, 441; Energy Environ. Sci. 2018, 11, 1819.

Response: Thank you for your helpful suggestions. We have added the references at the right place.

Comment 2: In the Introduction section, the authors mention several strategies for improving the HER performance (lines 41-45). However, they miss an important one: the substrate/support effect. In some cases, loading catalysts on a proper substrate/support can alter the electronic structure of catalysts and thereby substantially boost the catalytic performance, like that demonstrated in the above EES paper.

Response: Thank you for your very helpful suggestions. We have added this strategy (substrate/support effect) in the introduction part.

Comment 3: The overpotential needed to deliver a current density of 10 mA cm⁻² is only an indicator of apparent catalytic activity and is heavily dependent on the catalyst loading. The authors are recommended to specify the loading mass of Co-substituted Ru NSs and all other control catalysts. It is unfair if these catalysts are compared with different loading mass. Alternatively, the authors may consider optimizing the catalytic activity by tuning the loading mass, and making comparison based on the optimal activity. Furthermore, they need to elaborate how the TOF values are calculated either in the main text or in the Methods section. Are they calculated based on the catalyst mass or ECSA?

Response: We appreciate the suggestions from this reviewer. In our electrochemical test, the loading amount of the Co-substituted Ru catalysts was 0.153 mg/cm². For comparison, the Ru catalysts, RuCo alloy and commercial Pt/C catalysts have kept the same loading amounts of 0.153 mg/cm² under the same test conditions. In fact, we optimize the catalytic activity of the Co-substituted Ru catalysts by tuning the loading mass before electrochemical test. As shown in Figure S8, the dependence of HER rate on the loading amount of the Co-substituted Ru can be observed. The optimal loading is 0.153 mg/cm² in 1.0 M KOH. When the loading weight is higher, a lower catalytic performance is obtained due to the slow mass transfer and high charge transfer resistance.

In our text, the TOF values can be calculated by the following equation: $TOF=I/(2Fn)$, where I represents the current, F represents the Faraday constant, n represents the number of active sites. Forming H₂ needs two electrons that leads to the factor of 1/2 in this equation. Here, the number of active sites can be qualified by the following equation: $n=Q_{Cu}/2F$, where Q_{Cu} represents the underpotential deposition (UPD) Cu stripping charge ($Cu\ upd \rightarrow Cu^{2+} + 2e^-$). As shown in Figure S11a, Co-substituted Ru electrode was polarized at 0.215, 0.210, and 0.205 V for 100 s to form the UPD layers, respectively. Under the polarization potentials of 0.215 V, there is only one oxidation peak around 0.41 V, which belongs to the underpotentially deposited mono- or submonolayer copper. When the potential was decreased to 0.210 and 0.205 V, another oxidation peak appeared, which can be attributed to the oxidation of bulk copper. Pt/C was also investigated using the same method. As a comparison, Pt/C was further characterized by the underpotential deposition of hydrogen (Figure S11b). We have added the above information in the revised text.

Figure S8. LSV curves of Co-substituted Ru catalysts at different loading weight in 1.0 M KOH. Scan rate: 5 mV s^{-1} .

Figure S11. Copper UPD in the presence of 5 mM CuSO_4 on (a) Co-substituted Ru, and (b) Pt/C catalysts.

Comment 4: The authors are recommended to provide chronopotentiometry data to demonstrate long-term stability of their catalysts.

Response: Thank you for your very helpful suggestions. The catalytic stability of the Co-substituted Ru NSs was evaluated by chronopotentiometry (CP) at a constant current density of 10 mA cm^{-2} in 1.0 M KOH electrolytes. As shown in Figure S13, the Co-substituted Ru catalysts show very good catalytic stability in alkaline media in the course of 72000 s

galvanostatic electrolysis, with little degradation. We have added this information in the revised text.

Figure S13. Chronopotentiometric curves of Co-substituted Ru catalysts recorded at a constant current density of 10 mA cm^{-2} in 1.0 M KOH.

Comment 5: There are some technical improvements that should be made, e.g.

Figure 1 caption: the authors should mention what the white circles indicate.

Figure 5: the sub-panels should be presented in alphabetical order, namely, a, b, c, d, instead of a, b, d, c.

Figures S5 and S8: No signals are visible from Co. Perhaps the authors need to provide a zoomed view of the spectrum. In principle, if the atomic ratio of Co is above 10%, Co K alpha and L alpha should have been observed.

Line 87, page 2: it should be “diffraction pattern”, but not “diffraction spectra”. “spectrum” usually refers to the variation of intensity as a function of wavelength/energy.

Line 99, page 2: it is better to insert “measurements” after “(XANES)”

The authors should specify how much iR compensation they have made.

Response: Thank you for your kind suggestions. The revisions are as follows:

- a) We remove the white circles from Figure 1C due to its unclear image. In order to gain more direct evidence to confirm the atomic structure of Co-substituted Ru nanosheets, EELS

measurements were also conducted. Please see more details in comment 1 from the Reviewer
3. We have added it in the revised text.

- b) We have revised the alphabetical order of Figure 5 (namely, a, b, c, d) according to the reviewer's comment.
- c) Thank you very much for pointing out this information. We carefully examined the EDX results, and found that there is something wrong with the atomic ratio between Ru and Co due to our carelessness. The actual Ru:Co atomic ratio determined by EDX was 94:6 based on our repeated experimental results. We have provided new EDX results in the revised text.
- d) Yes, the term "diffraction spectra" has been changed into "diffraction pattern".
- e) Yes, we have inserted the term "measurements" after "XANES".
- f) The specific percentage of the correction is 100%. We have added this information in the revised text.

Comment 6: There are many grammatical errors and typos that should be corrected, to just name a few

a) lines 23-24, abstract: it will read better if the sentence changes to "among all Pt-free electrocatalysts reported so far"

b) Line 27, abstract: it should be "new insight into the development of" or "new insight into developing"

line 39, page 1: it is better to replace "meet" with "address"

line 40, page 1: it should be "approaches to obtaining"

line 50, page 1: delete "very"

line 87, page 2: "are" should be corrected to "is". There is only one pattern in Figure S4.

Line 91, page 2: it is better to move "profile" to a place between "scanning" and "were"

Line 93, page 2: delete "Energy ...spectroscopy". It's already mentioned in line 90. "demonstrate" should be corrected to "demonstrates"

Line 112, page 2: "position" should be used together with "positive" or "negative", and "longer" should be used together with "distance" or "length"

Line 113, page 3: "highest" cannot be used together with "than"

Line 128, page 3: I would suggest replacing “all” with “many other”, unless the authors can really compare Co-substituted Ru with ALL Pt-free HER catalysts. This is also the case in line 158.

Line 147, page 3: replace “to” with “with”

Lines 178-179: not clear what this sentence means

Line 185: replace “However” with “Moreover”?

Line 190: “increase” should be corrected to “increasing”

Line 251: “electrodes” should be corrected to “electrode”

Response: Thank you for your thoughtful suggestions. We have carefully checked the manuscript and make the corresponding revisions, which are shown in the following Table.

Entry	Position	Initial Manuscript	Revised Manuscript
1	Lines 23-24	among all reported Pt-free electrocatalysts so far	exhibited top-level catalytic activity among all Pt-free electrocatalysts reported so far
2	Line 27	a new insight to develop	a new insight into the development of
3	Line 39, Page 1:	meet	address
4	Line 40, Page 1	approaches to obtain	approaches to obtaining
5	Line 50, Page 1	very	delete “very”
6	Line 87, Page 2	are	is
7	Line 91, Page 2	scanning were...	scanning profile were...
8	Line 93, Page 2	energy ...spectroscopy	delete energy...spectroscopy
9	Line 93, Page 2	demonstrate	demonstrates
10	Line 112, Page 2	longer	positive
11	Line 113, Page 3	highest	higher
12	Line 128, Page 3	all	many other
13	Line 147, Page 3	To	with
14	Line 178-179	indicating...	indicating stronger hydrogen binding abilities after Co atom substitution, which leads to the obstacle of thermodynamic H* desorption process in the Heyrovsky step
15	Line 185	However	Moreover
16	Line 190	increase	increasing
17	Line 251	electrodes	electrode

Response to Reviewer 3:

The main claim of this work is the integration of cobalt into ruthenium lattice as single atoms. This structure resulted in high HER activity in alkaline solution. In general, the performance is very good, but there are concerns of the claim on single Co atom in Ru sheets. Major revisions on single atom analysis are recommended.

Response: Thank you for your positive comments on our manuscript. We have revised our manuscript accordingly.

Comment 1: Synthesis: It is doubtful the synthesis procedure could achieve fine single atom catalysts, particularly that cobalt could be reduced and bound to ruthenium in the nanosheets using oleylamine/heptanol at 180 degree C. The added molar ratio 3:1 (Ru:Co) was high and the EDX also suggests a high cobalt amount of 11%. Based on the Fig. 1b and 1c, what should be the estimated Ru:Co ratio in the catalyst?

Response: We appreciate the suggestions from this reviewer. The added molar ratio of Ru:Co was 0.90:0.10 ($Ru=40/639*3=0.188$ mmol; $Co=5.2/257=0.02$ mmol) because there are 3 mol of Ru in $Ru_3(CO)_{12}$. Furthermore, we carefully examined the EDX results, and found that there is something wrong with the atomic ratio between Ru and Co due to our carelessness. The actual Ru:Co atomic ratio determined by EDX was around 0.94:0.06 based on our repeated experimental results (*Please see the EDX analysis shown below*). In this typical synthesis, cobalt cannot be completely reduced due to the weak reduction ability of oleylamine. Thus, the actual atomic ratio of Ru:Co was higher than that of the reactants, consistent with our previous studies (*Angew. Chem. Int. Ed.*, 2017, 56, 11971-11975).

Spherical aberration-corrected HAADF-STEM, Electron energy loss spectroscopy (EELS) measurements, and X-ray absorption fine structure (XAFS) are the most powerful tools for investigating the structure of Co-substituted Ru nanosheets in atomic level. According to your suggestions, the aberration-corrected HAADF-STEM, EELS as well as XAFS measuring techniques were carried out. The evidences are depicted as follows:

1) Spherical aberration-corrected HAADF-STEM analysis

To gain an insight into the crystal structures and surface conditions of Co-substituted Ru

nanosheets, spherical aberration-corrected STEM was conducted. In HAADF-STEM images, the brightness reflects electronic intensity of corresponding area. As shown in HAADF-STEM images (Figure 1c), the brighter areas indicate thicker atomic layers (Ru), and the darker areas are depressions (Co). Atoms with different atomic weights can be distinguished in the atomic-resolution HAADF images (*Science* 2008, 321, 1331; *Ultramicroscopy* 1979, 4, 101). In order to see more clearly, false color was used to increase the contrast for the different atoms. Overall, we could not find a large area of darkness from Figure c, indicating a homogeneous distribution of Co atoms in the nanosheets.

2) EELS analysis

Electron energy loss spectroscopy (EELS) is an important tool for the study of the electronic structure and chemical distribution of materials. When fast electrons in the beam of an electron microscope (EM) penetrate a thin specimen, some will lose their energy to inner-shell electrons. The amount of energy lost is dependent on the atomic number of the specific atom being imaged, which allows for a determination of chemical composition using EELS. Here, we used EELS to study the elemental distribution of Co in Co-substituted Ru nanosheets by collecting localized core-edge EELS spectra on the sub-nm scale.

Figure 1d shows the HAADF line scanning profile of Co-substituted Ru nanosheets. Figure 1e shows the simultaneously acquired EELS spectra of the Co K-edge taken from the three atomic column locations indicated in Figure 1d. We only found that the position 1 exhibits typical Co $L_{2,3}$ -edge signal; while we could not observe the similar signals from neighbouring atoms, indicating a highly monodispersed distribution of Co atoms in the nanosheets.

3) XAFS analysis

Figure 2a shows the XANES curves at Ru K-edge of Co-substituted Ru NSs. It is found that the Ru K-edge of the sample slightly shifts to lower energy in comparison to that of Ru foil reference, suggesting that Ru in Co-substituted Ru NSs gets some electrons from Co. From the XANES spectra of Co K-edge in Figure 2b, we found that the absorption threshold of Co-substituted Ru NSs is between those of Co foil and CoO, indicating that Co in Co-substituted Ru NSs is positive charged. In order to further investigate the interaction between Ru and Co, Fourier transform (FT) EXAFS analysis is performed at Ru K-edge (Figure 2c) and Co K-edge (Figure 2d), respectively. In FT-EXAFS spectra of Ru K-edge, there is no peak related to Ru-O scattering, and the peak at

2.39 Å (slightly shorter than 2.41 Å of Ru-Ru in Ru foil) is due to Ru-Ru/Co contribution. In FT-EXAFS curves of Co K-edge (Figure 2d), Co-O coordination is not observed and the peak at 2.39 Å results from Co-Ru interaction. It is necessary to mention that the position of Co-Ru peaks is obviously positive than that of Co-Co in Co foil (2.16 Å), indicating the absence of Co-Co scattering in Co-substituted Ru NSs. The above results further demonstrate that Co atoms are isolated into the Ru lattice.

In a word, we tried our best to confirm the atomic structure of Co-substituted Ru nanosheets. By using atomic resolution Aberration-corrected HAADF-STEM (confirming homogeneous distribution of Co atoms throughout the whole nanosheets) coupled with EELS (confirming Co atoms are isolated into the Ru lattice), and XAFS (further confirming the Co atoms are isolated by Ru) multimethod to study its structure, we believe that the Co atoms locate at the hexagonal close-packed Ru lattice positions and isolated by Ru atoms, demonstrating the monodispersing of Co atoms in Co-substituted Ru NSs.

Figure S5. EDX pattern of Co-substituted Ru nanosheets. The Ru:Co atomic ratio was around 94:6.

Fig. 1. Characterization of Co-substituted Ru NSs. Representative (a) HAADF-STEM image and (b) AFM image of Co-substituted Ru NSs. (c) Atomic resolution HAADF-STEM image of a single Co-substituted Ru nanosheet. False color was used to increase the contrast. Inset of Figure c is the surface model of the nanosheet. (d) Line scanning profile and (e) Background subtracted EELS spectrum of the Co-substituted Ru NSs acquired from Figure d. The red rectangle represents Co $L_{2,3}$ -edge signal. (f) EDS line scanning profile and EDX mapping of a single Co-substituted Ru NSs.

Comment 2: HAADF-STEM: It is hard to conclude the atoms circled in white are cobalt.

Response: Yes. In order to gain more direct evidence to confirm the atomic structure of Co-substituted Ru nanosheets, EELS measurements were also conducted. Please see more details in comment 1.

Comment 3: RuCo-alloy: Show TEM and EDX data for the comparison sample with the Ru:Co molar feed ratio of 1:3. There was not enough comparisons between the alloy and the single atom catalysts. To a certain extent, having higher concentrations of cobalt with comparable activity is potentially an improvement.

Response: Thanks to the review's valuable suggestion. We further increased the amount of Co in the RuCo catalysts and carefully studied their catalytic performance towards HER under the standard reaction conditions. We chose Ru:Co molar feed ratio of 1:3 as a precursor according to your suggestions. However, the resultant Ru:Co is more than 1:3 based on EDX results (Figure

S9a, Ru:Co atomic ratio was around 1:2), which is similar to our previous studies. Figure S9b shows the TEM image of the as-obtained RuCo₂ catalysts. It can be observed that the morphology of RuCo₂ catalysts containing the mixture of nanosheets and nanoparticles. As you suggested, the HER performance of RuCo₂ was conducted. Figure S10a shows the LSV curves of RuCo₂ catalysts. The RuCo₂ required an overpotential of 144 mV to achieve current density of 10 mA cm⁻², which is much higher than that of RuCo (40.0 mV), Ru/C (92.5 mV), and Pt/C (56.5 mV) catalysts, indicating its inferior HER activity. The HER kinetics of the above catalysts were also calculated via corresponding Tafel plots. As shown in Figure S10b, the Tafel slope of the RuCo₂ catalysts is 122 mV dec⁻¹, which is much higher than that of Ru/C (85 mV dec⁻¹), Pt/C (54 mV dec⁻¹) and RuCo alloy (39 mV dec⁻¹) catalysts. The above results show that the molar ratio of Co and Ru could greatly influence the catalytic activity of the HER. When increasing the amount of Co in RuCo catalysts, the HER performance would be decreased. The DFT studies revealed that the energy barrier of water dissociation has remarkably improved after increasing the amount of Co, being consistent with the experimental observation. We have added this information in the revised text.

Figure S9. (a) EDX analysis and (b) TEM image of RuCo₂ catalysts. The Ru:Co atomic ratio was 32:68.

Figure S10. (a) LSV curves of RuCo₂ electrocatalysts in 1 M KOH solution. (b) Tafel plots of the polarization curves in a.

Comment 4: In the Co K-edge FT EXAFS analysis, the Co-Ru fit is not convincing enough. There are other possible fits around $R \approx 2.7$ for the main peak (with a shoulder peak) of the red curve in Fig. 2.

Response: We appreciate the suggestions from this reviewer. The shoulder peak at about 2.0 Å is not a coordination peak (it is too long for Co-O, and too short for Co-Co or Co-Ru). Herein, the back scattering factor is nonlinear, and the main peak in R space is not with total Gaussian distribution after Fourier transform. As a result, a shoulder peak appears at 2.0 Å.

Comment 5: The details of electrochemical measurement should be added, including the loading amount of catalyst and scan rate. How long was the durability test?

Response: Thanks to the review's valuable suggestion. We added the details of electrochemical measurement in the experimental section. The revisions are as follows: The electrochemical measurements were conducted on a three-electrode system (Figure S6) controlled by a potentiostat (V3, Princeton Applied Research). The catalyst-modified glassy carbon (GC,) electrodes, graphite rod and saturated calomel electrode (SCE) were used as working electrode (geometric area of a working electrode: 0.196 cm²), counter electrode (CE), and reference electrode (The SCE was calibrated, Figure S7), respectively. Before test, GC substrate electrodes were polished with 0.3 and 0.05 mm Al₂O₃ slurry and then sonicated in ethanol and water each for several times to make it clean. For preparation of the electrode materials, 2 mg of catalyst powder was dispersed in isopropanol/distilled water (Milli-Q, volume ratio,1:1) mixed solution containing 0.05 wt % of Nafion. The mixture was ultrasonicated for about 2 h to generate a

homogeneous ink. After that, 15 μL of ink solutions was transferred onto the GC rotating disk electrode and then dried at room temperature. We optimize the catalytic activity of the Co-substituted Ru catalysts by tuning the loading mass before electrochemical test. As shown in Figure S8, the optimal loading of the catalysts was 0.153 mg/cm^2 in 1.0 M KOH. The hydrogen-saturated 1M KOH aqueous solutions were used as the electrolytes before each test. The scan rate was set to 5 mV s^{-1} for LSV measurements. All polarization curves were corrected for the iR compensation (the specific percentage of the correction is 100 %). The electrochemical tests were performed at room temperature.

The TOF values can be calculated by the following equation: $\text{TOF} = I / (2Fn)$, where I represents the current, F represents the Faraday constant, n represents the number of active sites. Forming H_2 needs two electrons that leads to the factor of 1/2 in this equation. Here, the number of active sites can be qualified by the following equation: $n = Q_{\text{Cu}} / 2F$, where Q_{Cu} represents the underpotential deposition (UPD, Figure S11) copper stripping charge ($\text{Cu upd} \rightarrow \text{Cu}^{2+} + 2\text{e}^-$).

As for durability test, the Figure 3d shows the linear sweep voltammetry (LSV) curves measured before and after 1000 cycles at a scan rate of 5 mV S^{-1} . After 1000 CV cycles, the polarization curve for Co-substituted Ru NSs presents a negligible difference of the overpotential. Moreover, the catalytic stability of the Co-substituted Ru NSs was also evaluated by chronopotentiometry (CP) at a constant current density of 10 mA cm^{-2} in 1.0 M KOH electrolytes, as shown in Figure S13. The Co-substituted Ru catalysts show very good catalytic stability in alkaline media in the course of 72000 s galvanostatic electrolysis, with little degradation. We have added this information in the revised text.

Figure S6. Photograph of the three-electrode system for the electrochemical test.

Figure S7. LSV curves of Pt in 1.0 M KOH solution (H_2 -saturated), used for calibration of the SCE with respect to RHE. Scan rate: 5 mV s^{-1} .

Figure S8. LSV curves of Co-substituted Ru catalysts at different loading weight in 1.0 M KOH.

Figure S11. Copper UPD in the presence of 5 mM CuSO_4 on (a) Co-substituted Ru, and (b) Pt/C.

Figure S13. Chronopotentiometric curves of Co-substituted Ru catalysts recorded at a constant

current density of 10 mA cm^{-2} in 1.0 M KOH.

REVIEWERS' COMMENTS:

Reviewer #1

The revision has certainly improved the paper.

I recommend publication after some additional amendments:

a) there are still some imperfections in the text, see e.g. the Introduction:

-Line 34: ... than that in alkaline media;

- Line 47: ... (23, 24, 26-28). Although ...

-Line 53: ... novel Pt-free electrocatalysts at the atomic level, ...

A thorough check of the entire text is recommended.

b) I would indicate with numbers 1-3) the equations a-c).

c) I would also suggest to clarify the nature of the "active sites" mentioned at the end of the "Methods" section, just before the acknowledgments. I understand that the Cu UPD occurs over the entire surface, including the Ru atoms. If this is the case, the obtained value of TOF is normalized over the entire surface and represents an average value. Or did I misunderstand? The authors should clarify this point and better discuss the experimental evidence and the conclusions.

Reviewer #2 (Remarks to the Author):

The authors have addressed all of my concerns. I think the present manuscript can be accepted for publication.

Reviewer #3 (Remarks to the Author):

The authors have corrected the main errors in the manuscript, and presented enough new evidence to address the raised questions. I have no further request.

Reviewer #1

The revision has certainly improved the paper.

I recommend publication after some additional amendments:

a) there are still some imperfections in the text, see e.g. the Introduction:

- Line 34: ... than that in alkaline media;

- Line 47: ... (23, 24, 26-28). Although ...

- Line 53: ... novel Pt-free electrocatalysts at the atomic level, ...

A thorough check of the entire text is recommended.

b) I would indicate with numbers 1-3) the equations a-c).

c) I would also suggest to clarify the nature of the “active sites” mentioned at the end of the “Methods” section, just before the acknowledgments. I understand that the Cu UPD occurs over the entire surface, including the Ru atoms. If this is the case, the obtained value of TOF is normalized over the entire surface and represents an average value. Or did I misunderstand? The authors should clarify this point and better discuss the experimental evidence and the conclusions.

Response: Thank you very much for you positive comments and thoughtful suggestions on our manuscript. We have revised our manuscript accordingly.

a) We have carefully checked the manuscript and make the corresponding revisions, which are shown in the following Table.

Entry	Position	Initial Manuscript	Revised Manuscript
1	Lines 23-24	among all reported Pt-free electrocatalysts so far	among all Pt-free electrocatalysts
2	Line 23	exhibited	exhibit
3	Line 34	that of in alkaline media	that in alkaline media
4	Line 39	the poor	poor
5	Line 40	commercialization use	commercial applications
6	Line 47	(23, 24, 26-28); Although	(23, 24, 26-28). Although
7	Line 53	electrocatalyst	electrocatalysts
8	Line 60	the best activity	top-level catalytic activity
9	Line 62	demonstrate	demonstrated
10	Line 88	neighbouring	neighboring
11	Line 99	were	are
12	Line 99	an atomic ratio	atomic ratios
13	Line 128	higher	highest
14	Line 128	that of the	that of
15	Line 153	overlapping	overlapped
16	Line 153	initial curve	initial one
17	Line 157	catalysts	catalyst
18	Line 160	in 1.0 M KOH	Delete “in 1.0 M KOH”
19	Line 188	leads to	should lead
20	Line 190	kinetics	kinetic

21	Line 201	and three	and three per unit cell
22	Line 218	reduced	reduce
23	Line 220	catalyst	catalysts
24	Line 221	theory	theoretical

- b) Thank you for your suggestions. “equations a-c)” has been replaced by “equations 1-3)”.
- c) Underpotential deposition (UPD) is the deposition of metal atoms onto an electrode surface in mono- or submonolayer quantities. Integration of the peak area of upd stripping can calculate the number of active sites via assumption of an adsorption ratio of a single Cu atom to each surface metal atom (here for Ru atom). Turnover frequency (TOF), is one of the important index to evaluate HER electrocatalysts. $TOF=I/Q_{Cu}$ ($TOF=I/2Fn=I/(2F\times Q_{Cu}/2F)$). Here the TOF are closely related with I and Q_{Cu} values. Introducing of Co atom into Ru lattice can greatly enhance the catalytic activity of HER. The further studies suggested that single Co atom substitution significantly reduce energy barrier of water dissociation, leading to the outstanding HER performance. We have discussed this in the manuscript. Many previous reports have used this method to investigate turnover frequency (TOF) of the catalyst. We have added the corresponding references at the right place.

References:

19. Mahmood, J. et al. An efficient and pH-universal ruthenium-based catalyst for the hydrogen evolution reaction. *Nat. Nanotechnol.* **12**, 441-446 (2017).
35. Trasatti, S. & Petrii, O. Real surface area measurements in electrochemistry. *Pure appl. chem.* **63**, 711-734 (1991).
36. Ohyama, J. et al. Size Specifically High Activity of Ru Nanoparticles for Hydrogen Oxidation Reaction in Alkaline Electrolyte. *J. Am. Chem. Soc.* **135**, 8016-8021 (2013).
37. Green, C. L. & Kucernak, A. Determination of the Platinum and Ruthenium Surface Areas in Platinum-Ruthenium Alloy Electrocatalysts by Underpotential Deposition of Copper. I. Unsupported Catalysts. *J. Phys. Chem. B.* **106**, 1036-1047 (2002).